# Efficient Transformation of Somatic Embryos and Regeneration of Cork Oak Plantlets with a Gene (*CsTL1*) Encoding a Chestnut Thaumatin-Like Protein

**DOI:** 10.3390/ijms22041757

**Published:** 2021-02-10

**Authors:** Vanesa Cano, Mᵃ Teresa Martínez, José Luis Couselo, Elena Varas, Francisco Javier Vieitez, Elena Corredoira

**Affiliations:** 1Instituto de Investigaciones Agrobiológicas de Galicia (IIAG), Avd Vigo s/n, 15705 Santiago de Compostela, Spain; vanesa.cano.lazaro@iiag.csic.es (V.C.); temar@iiag.csic.es (Mᵃ.T.M.); jvieitez@iiag.csic.es (F.J.V.); 2Edificio de Ciencias Experimentais, Bloque A, Campus Universitario, Universidad de Vigo, 36310 Vigo, Spain; jlcouselo@semillasfito.com; 3Estación Fitopatolóxica do Areeiro (EFA), Subida a la Robleda s/n, 36153 Pontevedra, Spain; elenav@promiva.es

**Keywords:** *Agrobacterium tumefaciens*, in vitro *Phytophthora cinnamomi* tolerance test, oak decline, *Quercus suber*, pathogenesis related proteins, vitrification procedure, somatic embryogenesis, thaumatin-like protein

## Abstract

We present a reproducible procedure for transforming somatic embryos of cork oak with the *CsTL1* gene that codes for a thaumatin-like protein, in order to confer tolerance to *Phytophthora cinnamomi*. Different concentrations/combinations of the antibiotics carbenicillin and cefotaxime, as bacteriostatic agents, and kanamycin, as a selective agent, were tested. A lethal dose of 125 mg/L kanamycin was employed to select transgenic somatic embryos, and carbenicillin was used as a bacteriostatic agent at a concentration of 300 mg/L, which does not inhibit somatic embryo proliferation. The transformation efficiency was clearly genotype-dependent and was higher for the TGR3 genotype (17%) than for ALM80 (4.5%) and ALM6 (2%). Insertion of the transgenes in genomic DNA was confirmed by PCR analysis, whereas expression of the *CsTL1* gene was evaluated by semi-quantitative real-time PCR (qPCR) analysis. A vitrification treatment successfully cryopreserved the transgenic lines generated. The antifungal activity of the thaumatin-like protein expressed by the gene *CsTL1* was evaluated in an in vitro bioassay with the oomycete *P. cinnamomi*. Of the eight transgenic lines analyzed, seven survived for between one or two times longer than non-transgenic plantlets. Expression of the *CsTL1* gene and plantlet survival days were correlated, and survival was generally greater in plantlets that strongly expressed the *CsTL1* gene.

## 1. Introduction

Cork oak (*Quercus suber* L.) forests have been exploited over the years for cork production, which is of great economic interest. Cork is the outer bark of the tree and is harvested periodically, usually every 9–12 years, depending on the region [1]. This material is considered an elegant, high-quality product, due to its low weight, water resistance, elasticity and durability. Currently, 80% of global cork exports originate from Portugal and Spain, where the cork industry is one of the most technologically advanced, with high investment in Research & Development [2]. Manufacture of wine bottle stoppers is the main application for cork [1]. Cork is also used as a building material because it is a good thermal and acoustic insulator. In addition, cork oak acorns are used to fatten the Iberian pig, from which high-quality cured ham is obtained. However, this production system is at risk due to heavy tree mortality across the southwest of the Iberian Peninsula, due to a syndrome known as “La seca” in Spanish, “A secca” in Portuguese and “cork oak decline” in English. This disease has affected millions of trees throughout its range [3]. *Phytophthora cinnamomi* Rands, considered a hemibiotrophic oomycete [4], has been identified as the possible main cause of the disease [5,6,7]. *Diplodia mutila* Fr. apud Mont. (previous: *Botryosphaeria stevensii*) and *Biscogniauxia mediterranea* de Not. (previous: *Hypoxylon mediterraneum*) are considered secondary agents of oak disease [8,9,10]. Moreover, cork oak is affected by other pests such as the beetle *Coraebus undatus*, which excavates galleries in the tissue and greatly reduces the quantity and quality of the cork [11].

Cork oak decline is a very complex process and it is not easy to find solutions to restore the affected ecosystems [12]. Conventional breeding programs would take a long time to produce tolerant cork oak trees. On the other hand, there are currently no effective chemical treatments to control diseases caused by oomycetes such as *P. cinnamomi*. One promising possibility is the use of plant biotechnology through genetic transformation. The application of genetic engineering technology in tree physiology and biotechnology has been directed towards improving growth rates, wood properties and quality, pest resistance, stress tolerance and herbicide resistance. This is driving forestry to enter a new era of productivity and quality. Specific genes for resistance to oak decline have not yet been identified, but it is expected that in the coming years genes directly related to oak diseases will be identified in order to facilitate production of resistant trees [12]. The most commonly used approach to increase pathogen tolerance in plants is by overexpression of antimicrobial peptides, pathogenesis-related proteins (PR proteins) and proteins involved in the production of antimicrobial metabolites [13]. PR proteins are produced during the hypersensitive response (HR) and during systemic acquired resistance (SAR), and they act as part of the natural plant defence against pests and pathogens [14,15]. Seventeen classes of PR proteins have been described in trees and include the glucanases, chitinases and thaumatin-like proteins (TLPs) [16]. TLPs, which belong to the PR-5 family, are polypeptides of about 200 amino acid residues characterized by low molecular weight (less than 35 kDa) [17]. In order to defend the plant, TLPs can form transmembrane pores on fungal plasma membranes, thus promoting osmotic rupture, and/or they can inhibit hyphal growth and spore germination [16]. From a biotechnological viewpoint, the overexpression of thaumatin-like proteins has increased pathogen tolerance in several species such as bent grass [18], strawberry [19], wheat [20], grapevine [21] and cassava [22].

Some PR proteins with antifungal activity have been identified in recent years in European chestnut trees [23,24,25] belonging to the same family as cork oak. These proteins include the 23 kDa thaumatin-like protein, denoted *CsTL1,* which was purified from mature cotyledons of European chestnut and displayed in vitro antifungal activity against *Trichoderma viride* and *Fusarium oxysporum*. The *CsTL1* protein has also shown synergic effects with endochitinases and endoglucanases [25]. As this protein is part of the system of defence against microbial growth in chestnut, overexpression of this PR-protein in cork oak by using *Agrobacterium tumefaciens* is of potential value.

Genetic transformation of cork oak has been achieved in somatic embryos by using marker genes [26,27,28,29] as well as a gene that confers herbicide resistance [30], although the herbicide resistance has only been tested at laboratory level, as plantlet conversion was not reported. Despite this progress, genetic transformation in cork oak is strongly influenced by the genotype of embryogenic lines as few genotypes have been transformed. Genotype is considered the main factor for genetic transformation, and protocols defined for one clone or embryogenic line are often not appropriate for other clones or embryogenic lines [31]. Moreover, regeneration of plantlets from transformed somatic embryos of cork oak is difficult and the reports available do not provide details about conversion values [27]. These previous findings suggest the need for specific studies in cork oak to fine-tune transformation protocols, as an efficient protocol for cork oak genetic transformation of somatic embryos would be very beneficial for cork oak improvement programs.

The main objectives of the present study were (i) to develop an effective, reproducible procedure for transforming three different cork oak embryogenic lines, (ii) to produce cork oak trees expressing the *CsTL1* gene coding for a thaumatin-like protein and (iii) to evaluate the tolerance/resistance of these trees to *P. cinnamomi*.

## 2. Results

### 2.1. Effects of Antibiotics on Somatic Embryo Proliferation

To determine the lethal dose of kanamycin (Kan), cork oak somatic embryos isolated from two different embryogenic genotypes (ALM80 and TGR3) were cultured on medium containing different concentrations of Kan (Figure 1). The values of both parameters evaluated (i.e., the percentage of explants with secondary somatic embryos (SSEs) and the number of bipolar SSEs per dish) decreased as the concentration of Kan increased, although the toxic effect was more evident in the second of these parameters. In genotype ALM80, Kan concentrations higher than 75 mg/L severely affected the proliferation of somatic embryos (Figure 1a). In genotype TGR3, Kan produced a similar trend on the number of bipolar SSE as in ALM80, although the percentage of explants with SSE appeared to have been less affected by the concentration (Figure 1b). Kan concentrations of 125 mg/L Kan or higher completely inhibited the formation of bipolar SSEs (Figure 1).

The effect of the bactericidal antibiotics carbenicillin (CB) and cefotaxime (CF), used to prevent overgrowth of *Agrobacterium*, was also evaluated (Table 1). These antibiotics and concentrations did not significantly influence the percentage of explants with SSEs. By contrast, the number of bipolar SSEs per dish was affected by the antibiotic used as the bacteriostatic agent (Table 1). In both embryogenic lines, CB (300 mg/L) improved embryo proliferation, especially in genotype ALM80. The number of bipolar SSEs in this line was significantly (*p* ≤ 0.05) higher with CB 300 mg/L than in the control without antibiotics. By contrast, the corresponding number was lower in media with the same concentration of cefotaxime. Combinations of CB and CF also reduced the number of bipolar SSEs relative to the antibiotic-free medium. Kan (125 mg/L) was therefore chosen as the selective agent, while CB (300 mg/L) was chosen as the bacteriostatic agent for the transformation experiments with cork oak somatic embryos.

### 2.2. Genetic Transformation of Somatic Embryos

After cocultivation and culture on selective medium, target explants gradually turned brown, and evident signs of necrosis appeared after only two weeks of culture. Newly emerging somatic embryos or embryogenic structures began to appear in totally necrotic explants after 4 weeks of culture in the case of genotype TGR3, whereas a period of at least 8 weeks was required for genotypes ALM6 and ALM80. The experiment was completed after 10 weeks of culture on selective medium. The percentage of Kan-resistant explants was significantly (*p* ≤ 0.05) influenced by the genotype, and the highest percentages of Kan-resistant explants were obtained with TGR3 genotype (17%), followed by ALM80 (10.5%) and ALM6 (5.5%) (Figure 2). All explants cultured in the absence of the *Agrobacterium* suspension but including antibiotics (selective medium) became necrotic and did not survive (negative control).

Kan-resistant explants (Figure 3a,b) were isolated from the initial explants and sub-cultured on selective medium with 150 mg/L Kan, to confirm their capacity for embryogenic growth. The transformation efficiency (TE), defined as the percentage of initial explants that developed green fluorescent embryogenic cultures, was assessed four weeks later. Green fluorescent explants were observed in the three genotypes tested, and this parameter was also significantly (*p* ≤ 0.05) influenced by the genotype. The TE was highest in line TGR3 (17%) and was much lower in lines ALM80 (4.5%) and ALM6 (2%) (Figure 2). Expression of the fluorescent protein simplified and improved evaluation of the transformation events relative to the *GUS* assay used in previous protocols in cork oak transformation (Figure 3c,d). Overgrowth of *Agrobacterium* was not observed with CB 300 mg/L after 14 weeks of culture.

To ensure that each transformed line was derived from a single transformation event, only one embryo at cotyledonary developmental stage was isolated from each green fluorescent explant. Each isolated embryo was independently proliferated by repetitive embryogenesis on selective medium to establish the different cork oak transgenic embryogenic lines. A total of 30 cork oak transgenic lines were established with this protocol: 19 of these lines were derived from genotype TGR3, 7 were derived from genotype ALM80 and 4 were derived from genotype ALM6. Rapid secondary embryo proliferation was attained in all the transgenic lines established. Somatic embryos were morphologically indistinguishable from non-transgenic embryos and they developed into mature embryos through the normal developmental stages: globular, heart, torpedo and cotyledonary stages.

### 2.3. Molecular Analysis of Transgenic Lines

A total of 18 transgenic lines, corresponding to 60% of GFP positive lines obtained, were analyzed by PCR. The presence of the three transgenes (*NPTII, EGFP* and *CsTL1*) was first detected by PCR in all putative transgenic lines evaluated and in the positive control (plasmid), but not in the non-transgenic embryogenic lines (Figure 4). In addition, the presence of *CsTL1* gene was identified in both transcriptional directions (Figure 4). 

To verify *CsTL1* transgene expression, the total RNA was extracted from early cotyledonary embryos of 18 PCR-positive transgenic lines and their non-transgenic counterparts and analyzed by semi-quantitative real-time PCR (qPCR). The *CsTL1* transcripts were detected in all the tested transgenic lines, although greater differences were found in the expression of *CsTL1* gene (Figure 5). In ALM80 and ALM6 genotypes, *CsTL1* expression was significantly (*p* ≤ 0.05) higher in the three transgenic lines evaluated (Figure 5a,b). In ALM80, the highest levels of expression were obtained in lines ALM80-tau 19 and ALM80-tau 20 (24.9- and 22.9-fold, respectively) (Figure 5b), while in the ALM6 genotype, line ALM6-tau 6 showed the highest levels of expression (29.2-fold) (Figure 5a). 

In the TGR3 genotype, 12 transformed lines were analyzed (TGR3-tau 2, tau 4, tau 5, tau 6, tau 9, tau 18, tau 21, tau 23, tau 34, tau 36, tau 42 and tau 45), together with the wt line (TGR3-wt). The lines in which gene expression levels were significantly (*p* ≤ 0.05) highest were TGR3-tau 21 (5.43-fold), TGR3-tau 5 (2.66-fold), TGR3-tau 6 (2.47-fold) and TGR3-tau 18 (2.4-fold) (Figure 5c). Lines TGR3-tau 34, TGR3-tau 45, TGR3-tau 23 and TGR3-tau 2 and TGR3-tau 9 also overexpressed the gene, but with no significant differences. Finally, three lines showed an expression level equal to or slightly higher than the wt line: TGR3-tau 4, TGR3-tau 36 and TGR3-tau 42.

Of the 18 lines analyzed, 15 overexpressed the *CsTL1* gene, representing 83.33% of all lines obtained. Expression of the endogenous gene *CsTL1* and of a close homolog, named *CsTL2*, was also detected in all wt lines, although variable. Significant differences in the levels of expression of the *CsTL1* gene inserted in the different genotypes were observed. Specifically, the TGR3-wt line showed higher basal levels of endogenous thaumatin expression than ALM6-wt and ALM80-wt, almost as high as the transformed lines of these genotypes (data not shown). This may be due to the differences in the endogenous thaumatin expression in each of the wt lines.

### 2.4. Cryopreservation of Somatic Embryos

The vitrification procedure was successfully used to cryopreserve transgenic somatic embryos of cork oak. The procedure was applied to all transgenic lines in which *CsTL1* expression was analyzed by qPCR. In all lines, survival rates were higher than 60%, and rates of 100% were even obtained in some lines (Online Resource 1A). In all transgenic lines, the embryo recovery rates were higher than those achieved in the non-transgenic lines, with the exception of ALM80-tau 13, ALM80-tau 20 and TGR3-tau 4 (Online Resource 1A). The presence of three transgenes was also confirmed by PCR analysis of DNA extracted from both uncooled and cryopreserved somatic embryos (Online Resource 1B). All 18 cork oak transgenic lines were then stored indefinitely in LN. 

### 2.5. Analysis of the Regeneration Ability of Transgenic Embryogenic Lines

In all transgenic lines evaluated, partial germination of somatic embryos with only root development was observed (Table 2). Conversion of the embryos into plantlets with simultaneous development of shoot and root also occurred in all transgenic lines evaluated (Table 2), although the conversion frequencies varied greatly depending on the genotype (13.9–33.3% for ALM6 transgenic lines, 2.8–8.3% for ALM80 transgenic lines and 5.6–36.1% for TGR3 transgenic lines). The percentage conversion was higher in non-transgenic lines than in transgenic lines, except for TGR3-tau 21 and TGR3-tau 36. Remarkably long shoots with several leaves were obtained in all transgenic lines, especially in lines derived from genotype TGR3 (Table 2). The transgenic plantlets appeared normal and healthy, and no morphological differences relative to non-transgenic plantlets were observed (Figure 3e,f). *GFP* expression was observed in roots, leaves and shoots isolated from transgenic plantlets (Figure 3g,h). Fluorescence was more intense in young, small leaves than in older, larger leaves, and it was easier to spot in the midrib and petioles than in blades. *GFP* expression was not detected in roots, leaves or shoots obtained from non-transgenic plantlets used as negative controls (Figure 3g,h).

For production of a constant number of more uniform plantlets (in terms of shoot and root development) for use in the tolerance assays, one shoot isolated from each transgenic line with the highest levels of *CsTL1* expression was used to establish axillary shoot cultures. Shoot cultures were proliferated by axillary bud proliferation, and the proliferation rates were analogous in shoots of both transgenic and non-transgenic origin (data not shown). Root development was observed in all transgenic and non-transgenic lines evaluated, with values ranging between 51.7% and 98.3% (Table 3). In the ALM6 genotype, the rooting rate was significantly affected (*p* ≤ 0.05) by the line, and the best results were obtained with the ALM6-tau 6 (93.3%) and in the wt line (86.7%) (Table 3). In the ALM80 genotype, rooting percentage was also significantly influenced (*p* ≤ 0.05) by the line, and the best results were obtained with the ALM80-tau 20 line (86.7%) and the non-transgenic line (75.0%). Finally, in genotype TGR3, rooting rates were also significantly influenced (*p* ≤ 0.05) by the line, with the highest values observed in lines TGR3-tau 21 and TGR3-tau 6 (98.3% and 91.7%, respectively), while the rooting percentage was lowest in the non-transgenic line (76.7%) (Table 3). The number of roots per shoot varied between genotypes and lines, and it was higher in TGR3 transgenic lines (4.7–5.6) than in ALM6 transgenic lines (2.3–4.6) and ALM80 transgenic lines (1.6–2.4) (Table 3).

### 2.6. In Vitro Test of Tolerance to P. cinnamomi

In order to determine the tolerance of transgenic lines of cork oak to *P. cinnamomi*, plantlets rooted as described above were infected, and the number of days that each line survived was recorded. The symptoms of infection progressed in a similar way in the three genotypes, with necrosis of the roots and the stem first observed, followed by yellowing and necrosis of the leaves (Figure 6). Plantlets were considered dead when all of the organs were necrotic. After 31 days, the survival rate differed significantly in transformed and non-transgenic plantlets (Figure 7a–c). In the ALM6 lines, both transgenic lines ALM6-tau 6 and ALM6-tau 1 survived significantly (*p* ≤ 0.05) longer (19.9 and 19.2 days, respectively) than the ALM6-wt line (11.1 days) (Figure 7a). Similarly, in ALM80, both transgenic lines evaluated survived significantly (*p* ≤ 0.05) longer (23.2 days for ALM80-tau 20 line, 18.1 days for ALM80-tau 19 line) than the ALM80-wt line (11.5 days) (Figure 7b). Finally, in TGR3, transgenic plantlets of three lines showed a significantly (*p* ≤ 0.05) greater survival capacity (22.3, 25.2 and 29.5 survival days for TGR3-tau 5, tau 6 and tau 21 lines, respectively) than the non-transgenic line (13.1 days) (Figure 7c). By contrast, line TGR3-tau 18 (10.2 days) survived for less time than the TGR3-wt plantlets (Figure 7c).

Comparison of the three genotypes showed that non-transgenic plantlets (ALM6-wt, ALM80-wt and TGR3-wt) survived for a similar length of time (11.1 days, 11.5 days and 13.1 days, respectively) (Online Resource 2). Regarding transgenic lines, the level of expression of the *CsTL1* gene was correlated with the percentage survival of lines TGR3-tau 21, TGR3-tau 6, TGR3-tau 5 and ALM80-tau 20, which survived significantly (*p* ≤ 0.05) longer than the other transformed lines (Online Resource 2). 

At the end of experiment, survival was highest in plantlets from the lines in which the level of gene expression was highest. Four plants of line ALM6-tau 1 and 2 plantlets of line ALM6-tau 6 survived, whereas four plantlets of line ALM80-tau 19 and two plantlets of line ALM80-20 were recovered alive. Finally, 6 plantlets of line TGR3-tau 5, 8 plantlets of line TGR3-tau 6 and 16 plantlets of line TGR3-tau 21 remained alive at the end of the infection experiment. None of the non-transgenic plantlets remained alive four weeks after the infection.

## 3. Discussion

This is the first report of the transformation of cork oak with a gene in order to improve the tolerance of the species to oak decline. Cork oak has been transformed with marker genes [26,27], but there is only one example of genetic transformation with a gene of agronomic interest, and plant regeneration from transgenic somatic embryos has not been reported [30]. Moreover, the effectiveness of these protocols is limited as their success is highly dependent on the genotype of the embryogenic line used.

Teixeira da Silva and Fukai [32] pointed out that the success of plant genetic transformation depends on a balance being obtained between various factors: the selective agent, the agent used to remove *Agrobacterium* and the process of regeneration from transformed cells. The tolerance of target explants to the antibiotic used in the selection process varies widely [33] and is highly dependent on the species, genotype and type and concentration of antibiotic [34]. A wide range of concentrations of kanamycin have been used, usually between 25 and 200 mg/L. Some species have a high level of tolerance to kanamycin, such as walnut [35] and olive [36], for which a concentration of at least 200 mg/L is necessary to inhibit the proliferation of embryogenic cultures. By contrast, low kanamycin concentrations have been used in the transformation of avocado [37] and tea [38], for which concentrations of respectively 50 mg/L or 75 mg/L were used. In the present study, the toxicity test showed that cork oak is moderately tolerant to kanamycin, inhibiting the proliferation of somatic cork oak embryos with kanamycin concentrations higher than 125 mg/L. Kan concentrations slightly lower than selected in the present paper have been used for transformation of cork oak [26] and pedunculate oak [39].

Carbenicillin and cefotaxime are the bacteriostatic antibiotics most commonly used for removal of *Agrobacterium* cells after cocultivation. As with kanamycin, it is necessary to assess whether antibiotics have a negative effect on the proliferation of embryogenic cultures [40]. In the present study, the addition of CB 300 mg/L improved SE proliferation in comparison with proliferation medium without antibiotics. This beneficial effect of carbenicillin has been observed in the somatic embryogenesis of papaya [41] and in shoot and root organogenesis of strawberry [42]. By contrast, the use of cefotaxime was ruled out, as it has a detrimental effect on the growth of cork oak SE. These results contrast with those previously obtained with cork oak, in which 500 mg/L of CF was used to eliminate *Agrobacterium* [26,27], although these studies did not evaluate the effect of different bacteriostatic antibiotics and their concentrations. In the aforementioned study, [27] we were not able to transform SE from lines ALM6 and ALM80, and we therefore decided to use the previously defined protocol for the transformation of SE from European chestnut with the *CsTL1* gene. Applying this protocol, we successfully achieved transformation of the three cork oak embryogenic lines, including the above mentioned lines ALM6 and ALM80. The increase in the co-cultivation period from 2 days [26,27] to 5 days and the use of CB instead of CF probably contributed to inducing the transformation of the three genotypes. Chauhan et al. [43] pointed out that the type of antibiotic used in the transformation strongly influences the transformation efficiency. Likewise, co-cultivation time is another factor that significantly affects the transformation frequency [44]. Generally, an increase in the time during which the explants are in contact with the bacteria improves the transformation efficiency [31]. For camellia, Mondal et al. [38] recommend the application of a 5-day cocultivation period in the transformation of somatic embryos. The same cocultivation period has been successfully used in the genetic transformation of European chestnut [45], eucalyptus [46] and holm oak [47].

Although the three genotypes evaluated were successfully transformed, the TE was affected by the genotype and was greater in the TGR3 genotype (17%) than in the ALM80 (4.5%) and ALM6 (2%) genotypes. Numerous observations highlight the effect of the genotype on transformation ability. In *Q. robur*, the authors of [39] achieved transformation with marker genes of the five lines evaluated, with the frequencies ranging between 2% and 6%, depending on the genotype. The genotype was also found to influence the transformation of holm oak SE, with transformation only obtained in two of the three lines evaluated, and TE also varied with the embryogenic line [47]. 

The presence of the three transgenes was confirmed by PCR analysis of all the cork oak transgenic lines evaluated, indicating that the system of two marker genes and the lethal dose of Kan used were suitable for preventing escapes. Overexpression of the *CsTL1* gene was confirmed in 15 of the 18 lines analyzed by qPCR, in comparison with three reference genes. Differences in gene expression levels were found between the different transgenic lines evaluated, even though gene transcription is regulated by the strong constitutive promoter CaMV35S. This is consistent with our previous results for the genetic transformation with *CsTL1* gene of other Fagaceae species such as European chestnut [48] and pedunculate oak [49]. Likewise, Zhang et al. [50] also observed notable differences in expression of oxalate oxidase gene between the different embryogenic lines generated in American chestnut. Such differences in the transgene expression between transgenic lines may be due to the position that the transgene occupies in the plant genome [51], the size and organization of the transgene [52], the presence of inverted copies or the incorporation of incomplete copies of DNA-T [18,53,54,55], or even by inactivation of the expression of the transgene [22]. 

*GFP* evaluation clearly simplified and improved the evaluation of the transformation events in real time, relative to the *GUS* assay [56]. The presence of necrosed tissues or green spots may interfere in the *GUS* assay, making longer periods of incubation or maintenance of the embryos in darkness necessary to prevent greening [27]; however, such difficulties do not occur with *GFP*. Cytotoxic effects in plant cells caused by *GFP* expression have been mentioned in previous studies [57], but the data obtained in the proliferation of embryos and plant regeneration, as well as the morphological appearance of the transgenic material, suggest that *GFP* gene expression is not toxic in cork oak. Similarly, deleterious effects were also not observed in the *GFP*-mediated genetic transformation of somatic embryos of several citrus cultivars [58], avocado [59] or Chinese chestnut [60].

Cryopreservation may be a reliable method that reduces the costs associated with maintaining embryogenic cultures [61]. In the present study, the development of an efficient and reliable protocol for cryopreservation by vitrification of cork oak transgenic material, with high rates of survival and embryo recovery, enabled long-term conservation of the transgenic lines. In the same way, cryopreservation has also been reported in regard to the conservation of non-transgenic somatic embryos of *Q. suber* [62] and transgenic material such as somatic embryos of European chestnut [48] and pedunculate oak [49] and shoot tips of birch [63] and aspen [64].

Plant regeneration from somatic embryos is considered one of the main bottlenecks in the application of somatic embryogenesis [65]. In the case of transgenic embryos, plant regeneration is often even more difficult [59,66,67,68], usually due to the continued cultivation in media with antibiotics [69]. However, in cork oak, regeneration was obtained in all 18-cork oak transgenic lines evaluated, and the plantlets exhibited normal morphology and growth. This finding contrasts with those mentioned in previous studies on genetic transformation of cork oak, in which information about plant regeneration is scarce or omitted [26,27,30]. The frequency of conversion of cork oak lines transformed with *CsTL1* varies considerably depending on the genotype. The effect of genotype on plant regeneration has been reported for non-transgenic embryos [70,71,72,73,74], as well as for transgenic somatic embryos [48,49,59]. In addition, during genetic transformation by *A. tumefaciens*, it is not possible to control either the number of gene copies that are inserted or their position in the plant genome [75]. This could explain the differences observed in the conversion of the different transgenic lines, even within the same genotype.

Due to the low conversion frequencies of some transgenic lines and high variation in root length and/or shoots, we decided to establish axillary bud cultures from a shoot derived from a germinating somatic embryo. Ballester et al. [76] indicated that this strategy could be used as complementary to germination and regeneration of plants from somatic embryos, enabling the production of unlimited and constant number of transgenic rooted plantlets that are more uniform in terms of root and shoot length. Such uniformity is necessary for *P. cinnamomi* tolerance tests, as it prevents discrepancies in the results due to differences in the physiological and morphological state of plantlets. 

It is difficult to compare the tolerance to *P. cinnamomi* obtained with the transgenic cork oak plantlets and with other previous reports due to the significant differences in the experimental procedures used. These procedures differ in species, pathogens, plant age, or type of test (in vitro or ex vitro, excised organs or entire plant). However, in most of these studies in which the thaumatin-like protein was overexpressed, a substantial improvement in tolerance to the pathogens attacking them was observed. For example, a sweet orange plant of age 9–12 months transformed with the PR5 gene, which encodes a TLP identified in tomato, showed a significant reduction in lesion development relative to control plants, and survival rates were higher than in control plants [77]. Pure protein extractions obtained from the leaves of tobacco plants that overexpressed a TLP isolated in *Pyrus pyrifolia* inhibited the growth of *Phytophthora parasitica* var. nicotianae, *Sclerotinia sclerotiorum*, *Phomopsis* sp. and *Alternaria* sp [78]. Similar enhancement of resistance was observed in transgenic banana plants overexpressing the rice TLP gene and showing significant tolerance to *Fusarium* [79].

In cork oak, *P. cinnamomi* tolerance in plantlets transformed with the *CsTL1* gene was evaluated using an in vitro assay. In vitro tolerance tests have been used in other woody species, to determine, for example, *P. cinnamomi* tolerance in avocado rootstocks [80] or European chestnut [81]. The in vitro test revealed that of the eight transgenic cork oak lines analyzed, seven survived more days after infection than the non-transgenic plantlets, four of them significantly. The transformed plantlets of genotypes ALM6 and ALM80 survived at least one week longer than their non-transgenic counterparts, whereas in the TGR3 genotype, transgenic plantlets of lines 5, 6 and 21 survived two times longer than the non-transgenic plantlets. This type of test appears useful for evaluating pathogen tolerance in species such as cork oak, in which there are difficulties in the acclimatization step, because in vitro testing is a rapid and economic method for preliminary evaluation of the tolerance of transformed lines when sufficient numbers of acclimatized plants are not available for field tolerance tests.

## 4. Materials and Methods

### 4.1. Plant Material

For transformation experiments, three cork oak embryogenic genotypes, named TGR3, ALM80 and ALM6, initiated from leaf explants derived from mature *Quercus suber* trees [82] were used. Embryogenic cultures were proliferated by repetitive embryogenesis with subculture at 6-week intervals in Petri dishes (diameter 9 cm) containing 25 mL of multiplication medium consisting of Schenk and Hildebrandt (SH) mineral salts [83], Murashige and Skoog (MS) vitamins [84], 3% sucrose (*w*/*v*) and 0.6% plant propagation agar (*w*/*v*) (Condalab, Torrejón de Ardoz, Madrid, Spain). After the pH was adjusted to 5.6, the medium was autoclaved at 115 °C for 20 min. Unless otherwise indicated, the embryogenic cultures were subjected to a 16-h photoperiod (provided by cool-white fluorescent lamps at a photon flux density of 50–60 μmol m^−2^·s^−1^) and 25 °C light/20 °C dark temperatures (standard conditions).

### 4.2. Binary Vector and Agrobacterium Strain

The chestnut gene encoding a thaumatin-like protein (*CsTL1*) purified from mature European chestnut cotyledons [25] was used in this study. The *CsTL1* gene was cloned into the pK7WG2D vector by Gateway system (Invitrogen, Carlsbad, CA, USA) as described by [48]. The T-DNA region of the vector also contains the neomycin phosphotransferase (*NPTII*), as a selectable marker gene driven by the nopaline synthase (Nos) promoter, and green fluorescence protein (*GFP*), as a reporter gene driven by the rol root loci D (rolD) promoter. The binary vector, designated pK7TAU, was transferred into *Agrobacterium tumefaciens* strain [85] EHA105 by the freeze-thaw procedure [86] and used in the transformation experiments.

Briefly, cultures of EHA105pK7TAU were initiated from a glycerol stock and grown in semi-solid Luria-Bertani (LB) medium consisting of 1% tryptone (*w*/*v*), 0.5% yeast extract (*w*/*v*), 1% NaCl (*w*/*v*), 1.5% agar (*w*/*v*) and 50 mg/L Kan (pH 7.0) [87] for 2–3 days at 28 °C in darkness. A single colony was placed in 2 mL of LB medium with 50 mg/L Kan and the suspension was cultured overnight at 28 °C with shaking (180–200 rpm), in darkness. One ml of the bacterial culture was inoculated into 600 mL of LB liquid medium supplemented with 50 mg/L Kan, and the suspension was grown at 28 °C at 100 rpm in darkness until reaching an optical density of 0.6 at a wavelength of 600 nm (OD_600_). The cells were pelleted by centrifugation (6500 rpm for 10 min at 10 °C), and the pellet was resuspended in 200 mL of MS [84] liquid medium supplemented with 5% sucrose (*w*/*v*) to produce the infection medium.

### 4.3. Determination of Phytotoxic Levels of Antibiotics 

To test the kanamycin tolerance of cork oak somatic embryos, groups of embryos (2–3 somatic embryos at globular or torpedo developmental stage) were inoculated into multiplication medium supplemented with different concentrations of Kan (0, 25, 50, 75, 100, 125 and 150 mg/L). In a separate study, the effect of different combinations of the bacteriostatic antibiotics carbenicillin (200 or 300 mg/L) and cefotaxime (200 or 300 mg/L) were also evaluated (see Table 1). All antibiotics were filter-sterilized and added to cooled autoclaved medium. Four Petri dishes with eight embryo groups per dish were used for each treatment, and the experiment was conducted twice. The embryos were cultured for 8 weeks, and the phytotoxic effect was then tested by measuring the percentage of explants forming secondary somatic embryos (SSEs) and the number of bipolar SSEs per dish.

### 4.4. Agrobacterium Transformation Procedures

Target explants consisting of groups of two or three somatic embryos at globular or torpedo developmental stage were used in transformation experiments. The groups of somatic embryos were dissected from embryogenic cultures of three embryogenic genotypes (ALM6, ALM80 and TGR3), 4 weeks after the last subculture, and were pre-cultured on multiplication medium for 1 day.

Pre-cultured groups of the three embryogenic genotypes were immersed in infection medium for 30 min and were then filter-dried and transferred to multiplication medium. Five days later, co-cultivated explants were washed for 30 min with sterilized water containing 300 mg/L CB, filter-dried and transferred to Petri dishes (10 groups per dish) containing selective medium consisting of multiplication medium supplemented with 300 mg/L CB and 125 mg/L Kan. Explants were cultivated under standard conditions with periodic refreshment of medium every two weeks. After 10 weeks, the percentage of Kan-resistant explants was recorded and explants with new somatic embryos were then transferred to fresh selective medium supplemented with 150 mg/L Kan for a further 4 weeks. At the end of this period (14 weeks in total), explants growing on selective medium were evaluated by *GFP*-specific fluorescence.

One hundred embryogenic groups were used for each embryogenic genotype evaluated (300 explants in total). Twenty uninoculated embryo groups were cultured on multiplication medium with or without antibiotics, as negative and positive controls, respectively.

### 4.5. Evaluation of GFP Expresion and Establishment of Transgenic Lines

After 14 weeks of culture in selective medium, *GFP* visual analysis of Kan-resistant somatic embryos was performed using Leica M205 FA epifluorescence stereomicroscope equipped with *GFP* filter (470/40 × nm excitation filter and 525/50 nm long-pass emission filter) (Leica, Wetzlar, Germany). Images were taken with a camera (Leica DSC7000T, Leica, Germany). The transformation efficiency was defined as the percentage of initial explants that produced green fluorescent embryogenic cultures. To establish the different cork oak embryogenic transgenic lines, for each green fluorescent explant, a somatic embryo at cotyledonary developmental stage was proliferated on selective medium. The somatic embryos were routinely maintained by repetitive embryogenesis with sequential subculture at 5–6-week intervals according to the previously defined conditions. 

### 4.6. Molecular Characterization of Transgenic Embryogenic Lines

#### 4.6.1. PCR Analysis

PCR analysis was used to confirm the presence of the transgenes *CsTL1*, *NPTII* and *EGFP*. Realpure Spin Plants and Fungi Kit (Durviz, Valencia, Spain) was used to extract genomic DNA from the somatic embryos of non-transgenic (wt) and putative transgenic embryogenic lines. Reactions were carried out in a 25 μL volume containing 200–300 ng of genomic DNA, 0.6 μM each primer, 200 μM dNTPs, 2.5 mM MgCl_2_, 1 × supplied polymerase buffer (Qiagen, Germany) and 1 U Taq DNA polymerase (Qiagen, Hilden, Germany). PCR analysis was carried out with specific primers (see primer sequences, programs and size fragments in Online Resource 3) in a MJ Mini™ thermal cycler (Bio-Rad, Hercules, CA, USA). Plasmid DNA from pK7TAU and DNA from non-transgenic somatic embryos were used as positive and negative controls respectively. PCR products were analyzed by gel electrophoresis on agarose 1.5% (*w*/*v*) gels stained with Red Safe (iNtRON Biotechnology, Jungwon-Gu, South Korea) and visualized under a UV transilluminator.

#### 4.6.2. Quantification of *CsTL1* Overexpression by Semi-Quantitative Real-Time PCR

Semi-quantitative real-time PCR was used to evaluate expression of the *CsTL1* gene. Cork oak somatic embryos at early cotyledonary developmental stage of putatively transgenic lines and of the non-transgenic lines were isolated to obtain the total RNA, by using the Qiagen RNeasy Plant Mini Kit (Qiagen, Hamburg, Germany) following the manufacturer’s recommendations. To eliminate DNA contamination, all RNA samples were treated with DNase I Digestion Set (Qiagen, Hamburg, Germany). First-strand cDNA was prepared from 1 µg total RNA using the Quanta cDNA synthesis kit (Quanta Biosciences, Gaithersburg, Maryland, USA). qPCR was performed in an optical 48-well plate with an Eco Real-Time PCR System (Illumina, San Diego, CA, USA) with a final volume of 15 µL per reaction. Each reaction mixture consisted of 1 x Maxima SYBR Green qPCR Master Mix (Thermo Scientific, Waltham, MA, USA), 200 nM of each primer (see primer sequences in Online Resource 3) and 1 µL of cDNA template. The following standard thermal profile was used for all PCR reactions: 40 cycles of 95 °C for 15 s and 60 °C for 1 min. Melting curves were obtained after one cycle of 95 °C for 15 s, 55 °C for 15 s and 95 °C for 15 s. The expression data were normalized by expression of three reference genes: tubulin (*TUB*) [88], actin (*ACT*) and polymerase elongation factor (*EF*) [89] (see primer sequences in Online Resource 3), previously selected on the basis of their stability during embryo development by applying the geNorm software [90]. For each gene, primer efficiency was evaluated using a standard curve. 

A total of 18 transgenic lines were analyzed: 12 PCR-positive transgenic lines from the TGR3 genotype, 3 PCR-positive transgenic lines from the ALM80 genotype and 3 PCR-positive transgenic lines from the ALM6 genotype. Three or four independent biological replicate samples were assessed for each transgenic line, and samples were added to the plates in triplicate.

The relative expression of the *CsTL1* gene was expressed as fold changes, by using the comparative Ct method [91]. EcoStudy™ software v5.0.4890 (Illumina, San Diego, CA, USA) was used for all calculations and normalizations.

### 4.7. Cryopreservation of Transgenic Lines

Non-transgenic lines and transgenic lines, in which *CsTL1* expression levels were previously analyzed, were stored in liquid nitrogen (LN). A vitrification-based protocol, described by Valladares et al. [62], was used for cryopreservation of the lines. For cryopreservation, groups of 2–3 somatic embryos at globular or torpedo developmental stage were isolated from transgenic and non-transgenic embryogenic lines. Explants were cultured on multiplication medium supplemented with 0.3 M sucrose for 3 days and then placed in cryovials (10 explants per vial) with 1.8 mL of plant vitrification solution (PVS2) [92] for 60 min at 0 °C. The explants were then immediately immersed in LN. After one month in LN, cooled explants were thawed in a water bath (42 °C) for 2 min and cultured in multiplication medium. A total of 30 groups of somatic embryos were used for each embryogenic line. Survival and embryo recovery rates were determined after 6 weeks of culture. The presence of genes in transformed and cryopreserved somatic embryos was confirmed by PCR analysis.

### 4.8. Analysis of the Regeneration Ability of Transgenic Embryogenic Lines

Plant regeneration ability was evaluated in those transgenic lines in which expression of the *CsTL1* gene was previously analyzed. Cotyledonary somatic embryos (≥5 mm) were isolated from cork oak transgenic and non-transgenic lines. The embryos were then transferred to 100 mL baby food jars containing 30 mL of multiplication medium and stratified at 4 °C in darkness for two months. The somatic embryos were then transferred to 500 mL jars containing 70 mL of germination medium, consisting of SH medium supplemented with 3% sucrose (*w*/*v*), 0.6% Plant propagation agar (*w*/*v*) (Condalab, Torrejón de Ardoz, Madrid, Spain), 0.025 mg/L 6-benzyladenine (BA) and 0.05 mg/L indole-3-butyric acid (IBA), and cultured under standard conditions for 8 weeks [93]. The germination response was determined by recording the number of somatic embryos with only root development or only shoot development (partial germination), and the number of somatic embryos that converted into plantlets (both shoot and root development ≥5 mm). The length of roots and shoots and the number of leaves per regenerated plantlet were also recorded. Each line was tested in six replicate jars, each containing six mature somatic embryos (36 embryos per treatment). 

To produce uniform plantlets (in terms of root and shoot length) for use in the tolerance assays, axillary cultures were established from plantlets derived from somatic embryo germination. For each transgenic line, shoots from a somatic embryo that converted into plantlet and that displayed the highest levels of expression of the *CsTL1* gene were isolated and multiplied by axillary shoot proliferation with subsequent rooting. Micropropagation by axillary bud proliferation was achieved by culturing shoots in a vertical position on Gresshoff and Doy medium (GD) [94] supplemented with 3% sucrose (*w*/*v*) and 0.7% agar Bacto (*w*/*v*) (Dickinson and Company, France), with two sequences of changes at 3-week intervals in a 6-week multiplication cycle, as follows: 0.2 mg/L BA for the first 3 weeks and 0.1 mg/L BA for the next 3 weeks. 

For rooting, elongated shoots (≥1.5 cm) were cultured for 24 h in rooting medium, consisting of GD medium (1/3 strength macronutrients) [94] supplemented with 3% sucrose (*w*/*v*), 0.7% agar Bacto (*w*/*v*) (Dickinson and Company, Le Pont-de-Claix, France) and 25 mg/L IBA, and subsequent transfer to rooting medium without auxin for culture for 6 weeks under standard conditions. The rooting rate and the number of roots per shoot were evaluated at the end of this period. For each transgenic line, five baby food jars with six shoots were used (30 explants in total) and the experiment was repeated twice.

### 4.9. P. cinnamomi Infection Bioassay

The UEX1 strain of *P. cinnamomi* used in the present study was isolated in 2008, in Valverde de Mérida (Badajoz, Spain), as described by [7], and the pathogenicity and virulence were demonstrated by [95]. The strain was maintained on V8-agar medium [96,97] at 25 °C in darkness and subcultured at 2-week intervals. The isolates were passaged through chestnut or avocado leaves two weeks prior to the first test, as a precautionary measure against loss of pathogenicity through continuous subculturing [96,97].

Discs of diameter 5 mm were isolated from the perimeter of a 1-week old *P. cinnamomi* culture and cultured with a 1/10 dilution of clarified V8 liquid medium for 24 h (25 °C), before sporangia production was stimulated by incubating the discs four times, for 30 min each time, with Chen and Zentmyer salt solution [96]. The discs were then incubated overnight in salt solution under blue light (25 °C) and sporangia growth was visually confirmed by stereomicroscopy. Zoospore release was then induced by incubation for 10–15 min at 4 °C.

Prior to infection, the rooted plantlets, obtained as described above, were transferred to tubes with 16.5 mL of liquid GD medium [94] and with a paper bridge. The lower part of the tube was kept away from the light to enhance root growth. For infection of plantlets, a single disc obtained as described above was inserted in the bottom of each tube, and the plantlets were then grown under standard conditions.

Plantlets were observed daily for 31 days to determine the number of days that the plantlets survived infection. Plantlets were considered dead when necrosis occurred in 100% of the organs, or if other signs of decay were evident on the surface (yellow or brown roots, shoots and leaves or falling leaves). Once a plantlet was considered dead, the organs were collected and placed on V8-agar medium for 48–72 h in darkness, at 25 °C, to confirm oomycete recovery. At the end of experiment, surviving plantlets (plantlets that were still growing and producing apparently healthy leaves) were collected and the most apical parts of each plantlet were recovered and transferred to GD medium [94] for proliferation by axillary budding.

Tolerance was evaluated in transgenic lines in which the highest levels of expression of the *CsTL1* gene were obtained. For each transgenic line, six plantlets were infected, and the experiment was conducted thrice (18 plants per line). Non-transgenic plantlets (n = 6) were also infected as negative controls.

### 4.10. Statistical Analysis

The SPSS statistical package 23.0 for Windows (Chicago, Illinois, USA) was used for all statistical analyses. The influence of the bacteriostatic agents on embryo proliferation (Table 1) was analyzed by one-way analysis of variance (ANOVA I). The effect of genotype on the genetic transformation parameters was determined by Chi-squared test (Figure 2). Statistically significant differences between transgenic lines and their wt counterparts in *CsTL1* expression (as determined by qPCR; Figure 5), in rooting ability of axillary shoots (Table 3) and in oomycete tolerance (Figure 6) were evaluated by ANOVA I, whereas plant regeneration was evaluated by the Krustal Wallis test (Table 2). 

## 5. Conclusions

In conclusion, here we have described a highly efficient and reliable *Agrobacterium*-mediated transformation procedure that enables generation of large numbers of transgenic cork oak embryogenic lines and plantlets from many independent transformation events. The study findings indicate that the type of bacteriostatic antibiotic, the co-cultivation time, pre-culture time and genotype are important factors regarding the success of transformation in cork oak. Moreover, overexpression of the *CsTL1* gene improved the cork oak tolerance to *P. cinnamomi* infection. This new procedure opens the way to genetic modification of cork oak, thus enabling the evaluation of new and more specific genes that may confer resistance to *P. cinnamomi* or that may improve other desirable traits such as drought tolerance or cork production.

## Figures and Tables

**Figure 1 ijms-22-01757-f001:**
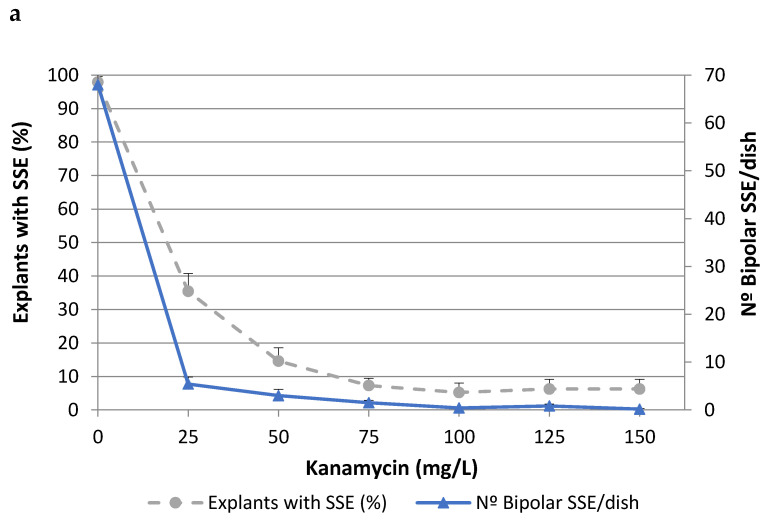
Effect of kanamycin concentration on percentage of explants with secondary somatic embryos (SSEs) and on the number of bipolar SSEs per Petri dish, in the genotypes ALM80 (**a**) and TGR3 (**b**). Vertical lines indicate the standard errors of mean values.

**Figure 2 ijms-22-01757-f002:**
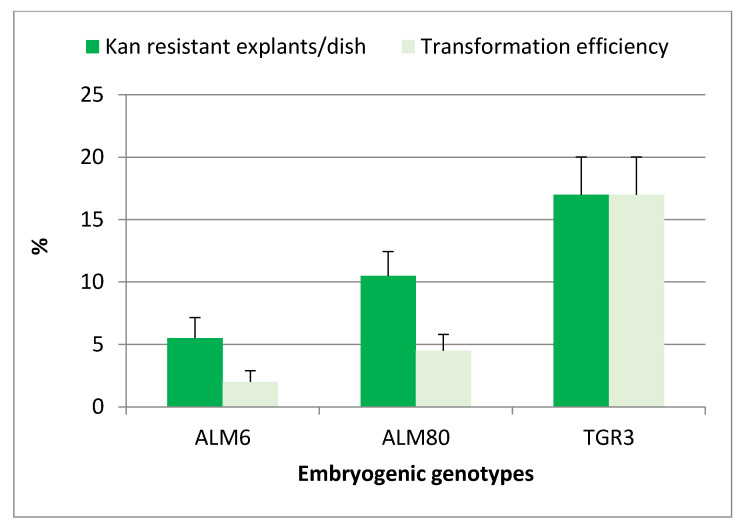
Effect of genotype on the percentage of kanamycin-resistant explants and transformation efficiency of cork oak somatic embryos transformed with strain EHA105pK7TAU. Ten Petri dishes were used for each genotype and each Petri dish contained 10 groups of somatic embryos. Vertical lines indicate the standard errors of mean values.

**Figure 3 ijms-22-01757-f003:**
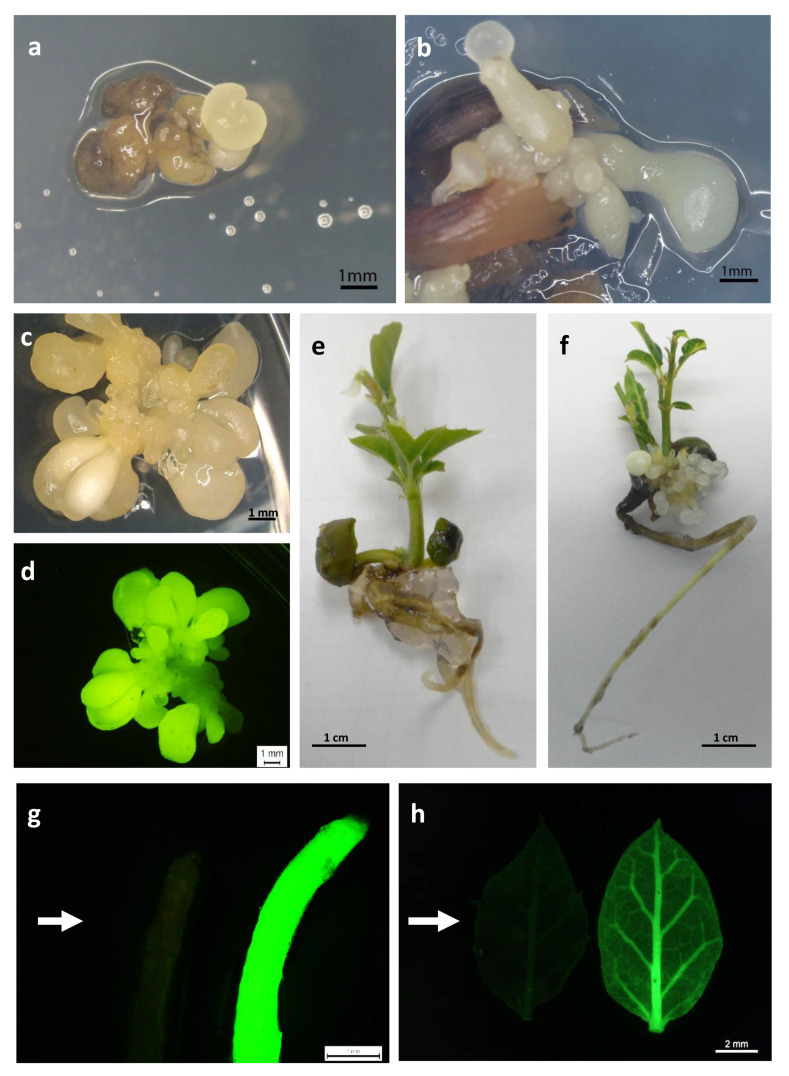
Different steps in the production of transgenic somatic embryos and transgenic cork oak plantlets. (**a**,**b**) Kan-resistant somatic embryos derived from genetic transformation of cork oak embryogenic genotypes ALM80 (**a**) and ALM6 (**b**) observed after 10 weeks in selective medium; (**c**,**d**) Transgenic cluster of somatic embryos visualized under white light (**c**) and green fluorescence protein (*GFP*) filter (**d**); (**e**,**f**) Plantlets obtained from germination of non-transgenic (**e**) and transgenic (**f**) somatic embryos after 6 weeks on germination medium; (**g**) *GFP* expression on a root of non-transgenic (left; arrow) and transgenic (right) plantlets, visualized in an epi-fluorescence stereomicroscope; (**h**) *GFP* expression on a leaf isolated from non-transgenic (left; arrow) and transgenic (right) plantlets visualized in an epi-fluorescence stereomicroscope.

**Figure 4 ijms-22-01757-f004:**
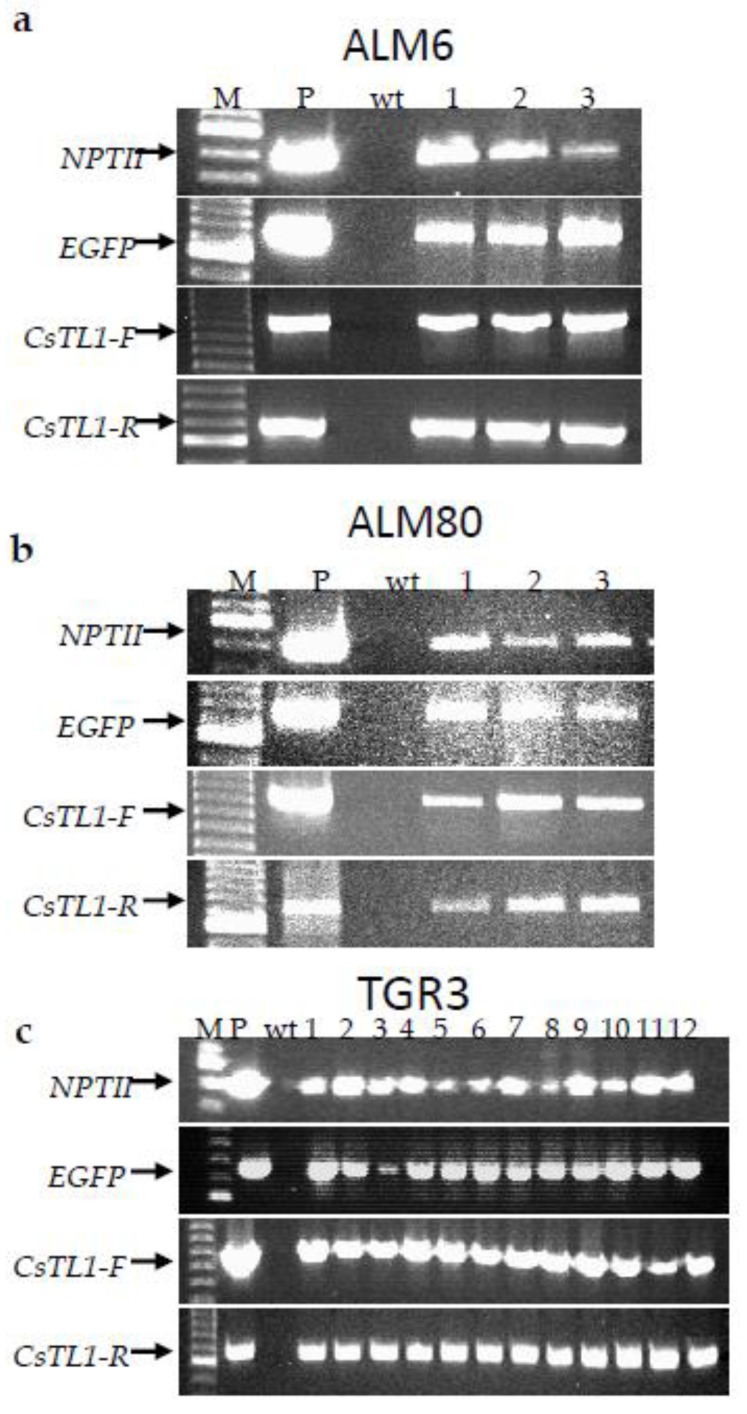
PCR analysis of transgenic lines to detect the presence CsTL1 (in both transcriptional senses), NPTII and GFP genes. M: Molecular ladder, ALM6 (**a**), ALM80 (**b**) and TGR3 (**c**). P: plas-mid (positive control), wt: non-transgenic somatic embryos of cork oak (negative control) and lanes 1–12: putative cork oak transgenic lines.

**Figure 5 ijms-22-01757-f005:**
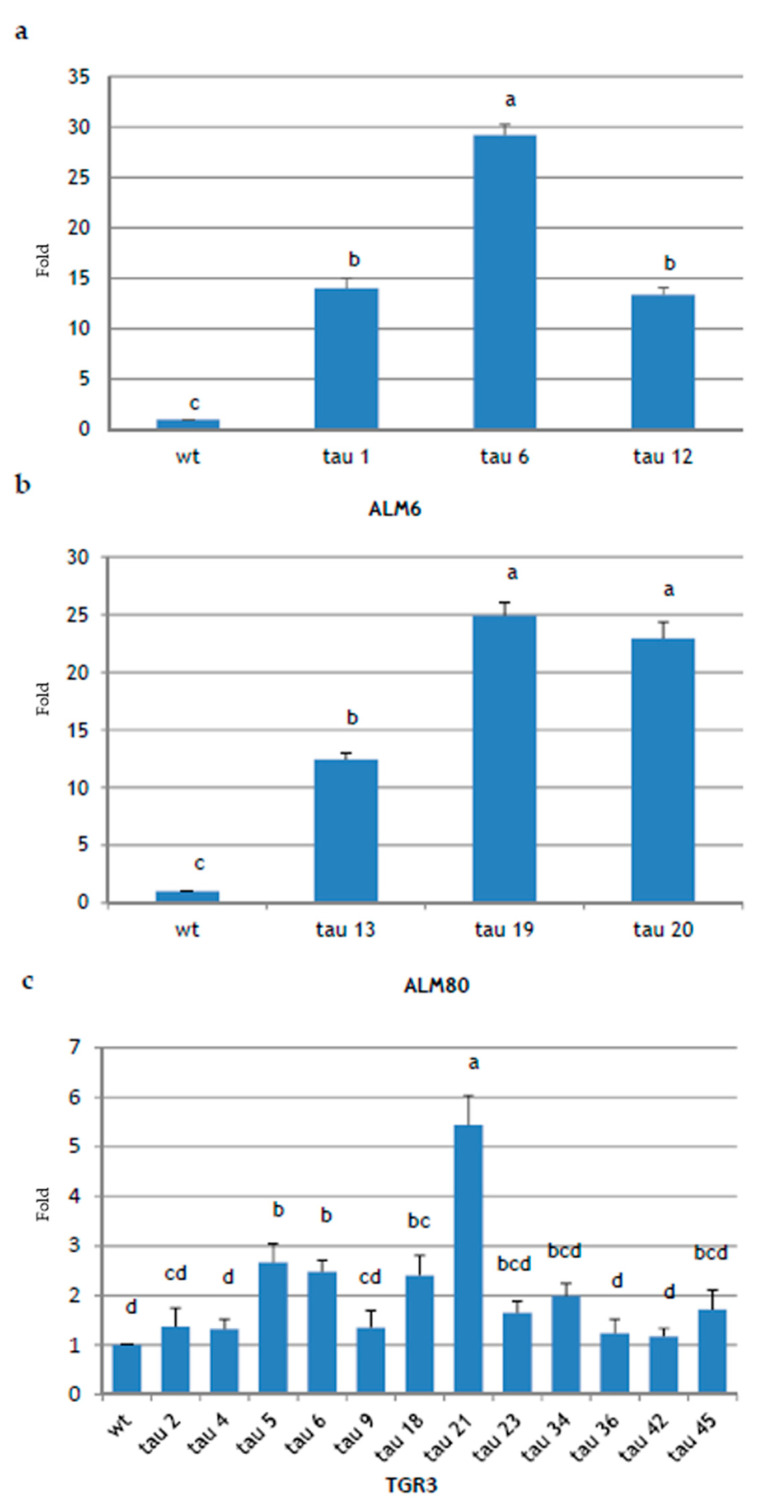
Expression analysis of *CsTL1* gene in somatic embryos of cork oak determined by qPCR. Total RNA was extracted from 18 independent transgenic lines and the corresponding non-transgenic lines (wt) of three cork oak embryogenic genotypes: ALM6 (**a**), ALM80 (**b**) and TGR3 (**c**). Values are mean ± standard error from at least three independent experiments, and vertical lines indicate the standard errors of mean values. Data analyzed by ANOVA I (*p* ≤ 0.05). Values in columns indicated with the same letter are not significantly different (*p* = 0.05, Duncan’s test).

**Figure 6 ijms-22-01757-f006:**
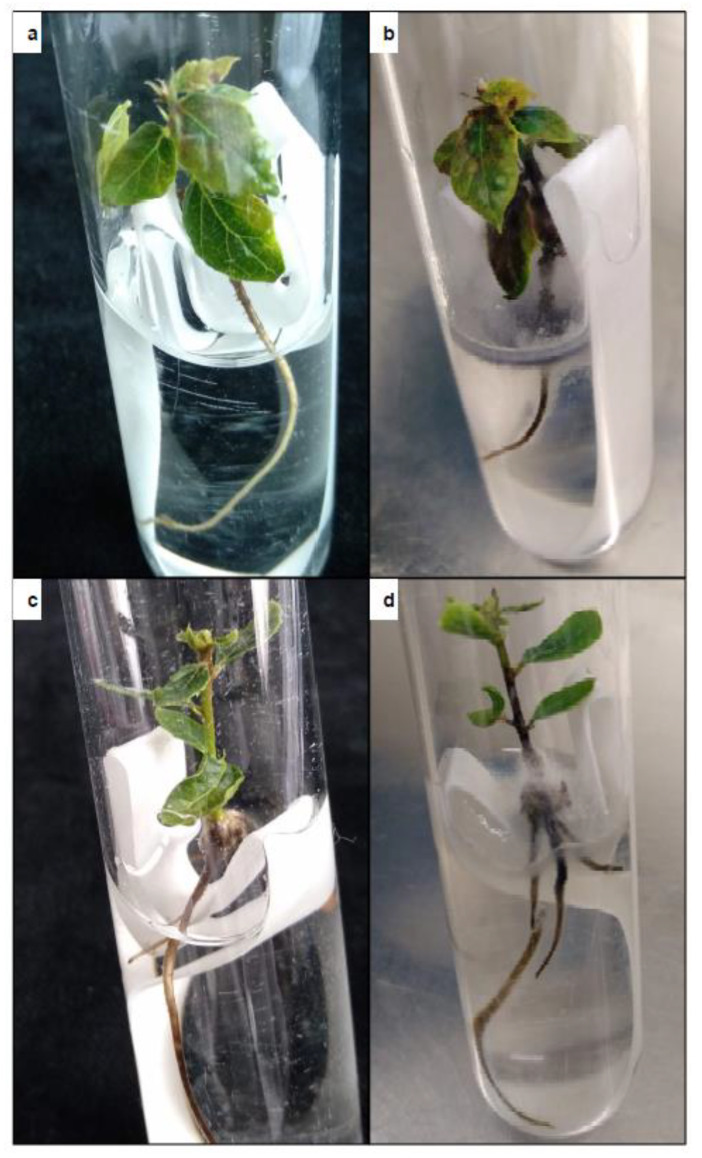
Morphological appearance and progress of infection in non-transgenic plantlets (**a**,**b**) and transgenic plantlets (**c**,**d**) of genotype ALM80 immediately after inoculation (**a**,**c**) and 15 days later (**b**,**d**). Diameter of the tube: 20 mm.

**Figure 7 ijms-22-01757-f007:**
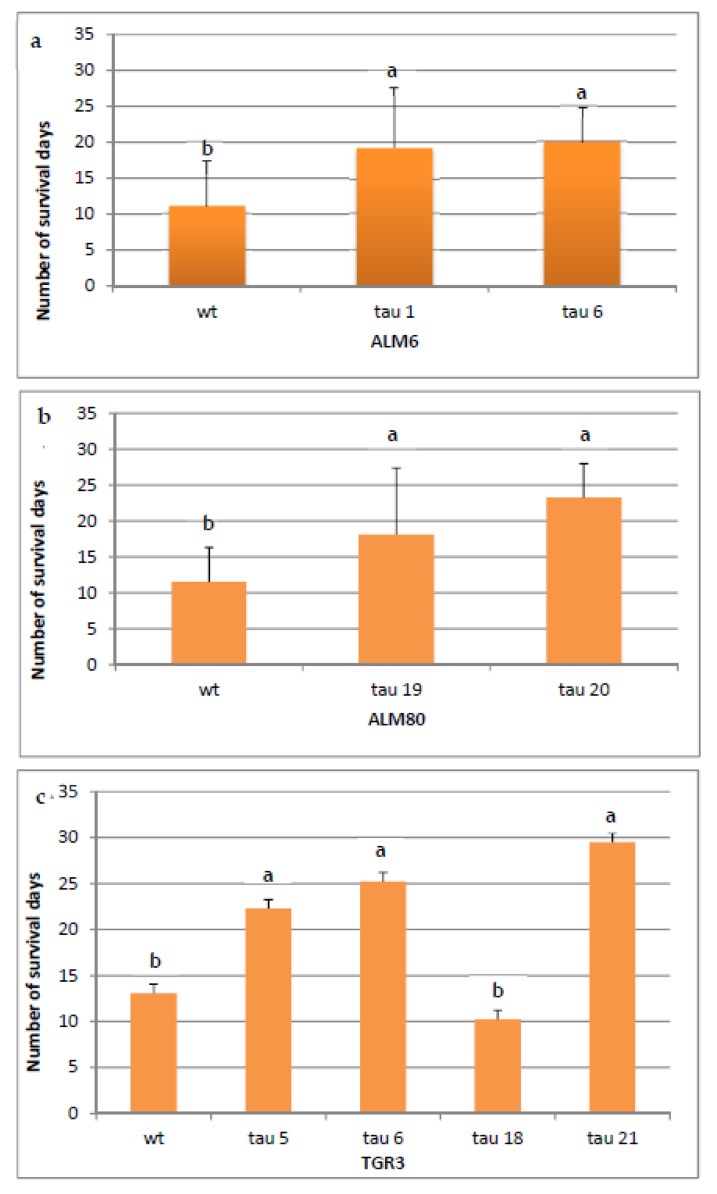
Survival rate of transgenic and non-transgenic (wt) cork oak lines derived from genotypes ALM6 (**a**), ALM80 (**b**) and TGR3 (**c**) 31 days after infection with *P. cinnamomi*. A total of 18 plantlets were evaluated for each transgenic and wt line. Vertical lines indicate the standard errors of mean values. Data were analyzed by ANOVA I (*p* ≤ 0.05). Values in columns indicated with the same letter are not significantly different (*p* = 0.05, Duncan’s test).

**Table 1 ijms-22-01757-t001:** Effect of different combinations/concentrations of bacteriostatic antibiotics on somatic embryo proliferation in cork oak embryogenic genotypes TGR3 and ALM80.

Antibiotics	Explants with SSE (%)		Number of Bipolar SSE/Dish	
(mg/L)	TGR3	ALM80	TGR3	ALM80
0	100.0 ± 0.0	98.4 ± 1.5	20.5 ± 0.7	71.3 ± 3.3
CF 200	100.0 ± 0.0	98.4 ± 1.5	19.6 ± 0.5	55.5 ± 3.7
CF 300	100.0 ± 0.0	100.0 ± 0.0	18.5 ± 1.0	48.5 ± 2.6
CB 300	100.0 ± 0.0	98.4 ± 1.5	20.5 ± 0.4	85.0 ± 9.7
CF 200 + CB 200	100.0 ± 0.0	100.0 ± 0.0	18.0 ± 0.8	68.3 ± 6.4
CF 200 + CB 300	100.0 ± 0.0	90.6 ± 2.9	18.3 ± 0.8	46.4 ± 3.2
F-test	ns	ns	ns	*p* ≤ 0.05

The values represent the mean ± standard error of eight replicates (eight explants each). SSE: secondary somatic embryos; CF: cefotaxime; CB: carbenicillin. The data were analyzed by one-way ANOVA. ns, not significant.

**Table 2 ijms-22-01757-t002:** Germination response with root only development, shoot only development and conversion into plantlets (root and shoot development) of cork oak somatic embryos from transgenic and non-transgenic (wt) lines after 2 months of cold storage and 8 weeks on germination medium.

Line	Root Only	Shoot Only	Conversion (Shoot + Root)			
	(%)	(%)	(%)	RL (mm)	SL (mm)	NL
ALM6-wt	29.2 ± 10.9	0.0 ± 0.0	41.7 ± 7.6	85.2 ± 14.7	21.9 ± 4.7	4.1 ± 0.9
ALM6-tau 1	25.0 ± 8.6	0.0 ± 0.0	19.4 ± 2.5	59.9 ± 18.0	18.5 ± 2.5	4.9 ± 1.6
ALM6-tau 6	41.7 ± 12.9	0.0 ± 0.0	33.3 ± 11.2	57.9 ± 12.6	19.2 ± 3.0	5.7 ± 1.0
ALM6-tau 12	55.6 ± 9.4	8.3 ± 3.4	13.9 ± 4.7	51.5 ± 8.0	30.6 ± 2.9	5.5 ± 0.2
H test	ns	*p* ≤ 0.05	*p* ≤ 0.05	ns	ns	ns
ALM80-wt	47.2 ± 7.3	2.8 ± 2.6	16.7 ± 3.9	93.2 ± 22.9	27.0 ± 14.0	6.2 ± 2.5
ALM80-tau 13	47.2 ± 14.4	0.0 ± 0.0	2.8 ± 2.5	58.0 ± 0.0	7.0 ± 0.0	3.0 ± 0.0
ALM80-tau 19	47.2 ± 7.3	0.0 ± 0.0	5.6 ± 5.1	105.0 ± 0.0	15.0 ± 0.0	4.0 ± 0.0
ALM80-tau 20	44.4 ± 10.2	0.0 ± 0.0	8.3 ± 5.2	149.3 ± 33.1	7.5 ± 1.0	6.0 ± 0.4
H test	ns	ns	ns	ns	ns	ns
TGR3-wt	22.2 ± 9.4	5.6 ± 3.2	18.1 ± 9.7	34.8 ± 9.3	21.5 ± 2.5	6.7 ± 1.1
TGR3-tau 2	27.8 ± 6.4	2.8 ± 2.5	13.9 ± 8.3	75.0 ± 12.3	28.2 ± 4.2	5.2 ± 0.5
TGR3-tau 4	56.9 ± 10.9	0.0 ± 0.0	16.7 ± 7.9	60.6 ± 10.5	22.8 ± 5.1	9.2 ± 1.1
TGR3-tau 5	38.9 ± 9.3	2.8 ± 0.0	16.7 ± 7.5	47.0 ± 24.4	8.7 ± 3.1	2.6 ± 1.0
TGR3-tau 6	47.2 ± 10.4	2.8 ± 2.6	11.1 ± 7.6	54.3 ± 3.4	16.3 ± 1.5	5.5 ± 0.2
TGR3-tau 9	43.1 ± 11.6	0.0 ± 0.0	5.6 ± 3.2	55.0 ± 8.2	5.0 ± 0.0	6.5 ± 1.0
TGR3-tau 18	37.5 ± 8.1	2.8 ± 2.6	5.6 ± 5.1	110.0 ± 0.0	22.0 ± 0.0	5.0 ± 0.0
TGR3-tau 21	23.6 ± 6.4	5.6 ± 3.2	34.7 ± 10.1	58.7 ± 23.2	12.2 ± 2.8	5.7 ± 0.8
TGR3-tau 23	8.3 ± 5.2	4.2 ± 3.8	15.3 ± 6.4	22.3 ± 3.0	19.2 ± 1.8	3.3 ± 0.5
TGR3-tau 34	41.7 ± 5.2	2.8 ± 2.5	16.7 ± 5.6	57.5 ± 13.0	11.9 ± 2.2	5.8 ± 0.5
TGR3-tau 36	38.9 ± 14.7	0.0 ± 0.0	36.1 ± 6.7	83.6 ± 15.1	8.6 ± 0.4	3.5 ± 0.4
TGR3-tau 42	36.1 ± 14.4	5.6 ± 3.2	11.1 ± 5.1	77.0 ± 10.6	13.3 ± 2.6	3.5 ± 0.2
TGR3-tau 45	34.7 ± 9.3	0.0 ± 0.0	11.1 ± 7.6	27.8 ± 1.1	19.3 ± 2.6	6.5 ± 1.0
H test	ns	ns	ns	ns	ns	ns

Each value represents the mean ± standard error of six replicates with six explants in each. Data were analyzed by the Kruskal-Wallis test (*p* ≤ 0.05). ns, not significant; RL, root length; SL, shoot length; NL, number of leaves.

**Table 3 ijms-22-01757-t003:** Adventitious rooting rates and number of roots of axillary shoots established from germinating cork oak somatic embryos of transgenic and non-transgenic lines (wt).

Line	Rooting (%)	Number of Roots
ALM6-wt	86.7 ± 5.2	2.8 ± 0.3
ALM6-tau 1	65.0 ± 6.9	2.3 ± 0.3
ALM6-tau 6	93.3 ± 3.5	4.6 ± 0.6
F-test	*p* ≤ 0.05	*p* ≤ 0.05
ALM80-wt	75.0 ± 4.3	1.9 ± 0.1
ALM80-tau 19	51.7 ± 3.7	1.6 ± 0.2
ALM80-tau 20	86.7 ± 4.6	2.4 ± 0.2
F-test	*p* ≤ 0.05	*p* ≤ 0.05
TGR3-wt	76.7 ± 5.9	5.1 ± 0.5
TGR3-tau 5	81.7 ± 5.0	5.6 ± 0.6
TGR3-tau 6	91.7 ± 3.5	5.2 ± 0.5
TGR3-tau 18	80.0 ± 5.7	4.7 ± 0.5
TGR3-tau 21	98.3 ± 1.6	5.0 ± 0.3
F-test	*p* ≤ 0.05	ns

Each value represents the mean ± standard error of six replicates (10 explants each). The data were analyzed by ANOVA I (*p* ≤ 0.05). ns, not significant.

## Data Availability

Data is contained within the article or Appendix A.

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
