# Peer review of "Efficient Transformation of Somatic Embryos and Regeneration of Cork Oak Plantlets with a Gene (CsTL1) Encoding a Chestnut Thaumatin-Like Protein"

_ijms, 2021, doi:10.3390/ijms22041757_

Round 1
Reviewer 1 Report
Dear Editor, Dear Authors,
The manuscript concerns an issue of genetic transformation of somatic embryos of cork oak with the CsTL1 gene to confer tolerance against Phytophthora cinnamomi. According to Authors this is the first report on transformation with gene of agronomic interest.
Although the topic scope seems narrow, the potential publication can be valuable for scientists interested in cork oak biology and its transgenic variants.
However, I regret to say that the manuscript cannot be publish in the present form.
I find the manuscript suitable for publication after major revision.
Main issues to be addressed:
- line 90 – “Moreover, the regeneration of plantlets from transformed somatic embryos of cork oak remains low” – could you please specify what “low” means?
- Could the Authors provide a little more details on parameters of Agrobacterium transformation (e.g. OD)?
- In section “Conclusion” the Authors wrote “This optimized procedure opens the way to genetic modification of cork oak”. Please, clarify what exactly was optimized referring to previous achievements/bottlenecks (transformation parameters, explant pre-treatment?).
- Is there any report on differences in cryopreservation of wild-type and transgenic embryos?
- Comments on references:
Could the Authors argument such a high number of references, and self-citation in particular?
If I am not mistaken, there is at least 13 papers (self-citations).
- Comments on the language:
I believe the MS needs some corrections since the authors did not avoid several minor grammar and punctuation mistakes (careful proofreading should solve the issue; however please, note I am not a native speaker); examples:
- lines 81: “overexpression the protein” – shouldn’t it be “overexpression of the protein” or protein overexpression?
- line 310: “…plant regeneration from transgenic somatic was not reported” – is it a correct sentence?
- line 313: I think there is an error.
- I believe, there is a typos in the title (thAumatin).
Sincerely yours,
reviewer
Author Response
Dear Editor and Reviewer,
We are grateful for the helpful corrections and suggestions made by the Reviewers to improve the quality of the original manuscript. Please find our comments to the Reviewer below. The new version of the manuscript has been modified accordingly.
With best wishes,
The authors
- Main issues to be addressed:
- line 90 – “Moreover, the regeneration of plantlets from transformed somatic embryos of cork oak remains low” – could you please specify what “low” means?
+ This sentence has been modified (see lines 79-80).
- Could the Authors provide a little more details on parameters of Agrobacteriumtransformation (e.g. OD)?
+This information has been included in the new version of the manuscript (see lines 618-625).
- In section “Conclusion” the Authors wrote “This optimized procedure opens the way to genetic modification of cork oak”. Please, clarify what exactly was optimized referring to previous achievements/bottlenecks (transformation parameters, explant pre-treatment?).
+ The Conclusions section has been modified to address these points (see lines 765-770).
- Is there any report on differences in cryopreservation of wild-type and transgenic embryos?
+To our knowledge, there are no reports on differences in the cryopreservation of wild-type and transgenic embryos.
- Comments on references:
- Could the Authors argument such a high number of references, and self-citation in particular? If I am not mistaken, there is at least 13 papers (self-citations).
+ We agree with the reviewer and it is probably true that the number of self-citations is rather high. However, in the last few years our research group has defined the first methods for the induction of somatic embryos and genetic transformation in three species in the family Fagaceae (oak, chestnut and holly oak), generally considered recalcitrant to in vitro culture and genetic transformation. The experience gained with these species has helped use to define an efficient protocol for cork oak, also previously considered recalcitrant to transformation. We have revised the self-citations included and have removed one of them.
Comments on the language:
I believe the MS needs some corrections since the authors did not avoid several minor grammar and punctuation mistakes (careful proofreading should solve the issue; however please, note I am not a native speaker); examples:
+The new version has been revised by a native English speaker.
- lines 81: “overexpression the protein” – shouldn’t it be “overexpression of the protein” or protein overexpression?
+Done.
- line 310: “…plant regeneration from transgenic somatic was not reported” – is it a correct sentence?
+This sentence has been clarified.
- line 313: I think there is an error.
+Yes. It has been corrected.
- I believe, there is a typos in the title (thAumatin).
+ Corrected.
Reviewer 2 Report
Dear authors,
I think this study has made a great achievement with a clear logic in writing and data presentation. I recommend accepting this paper after revision. However, it still can be improved. Please find my suggestions below:
- In title line 4, is it “thAumatin” ? or it should be Thaumatin. Please check it.
- In line 11, delete Here.
- In line 16, what does SE mean? Somatic embryos or secondary embryogenesis? Please verify it. If it has mentioned in the front text, please put the abbreviation when you write in the first time.
- In the abstract, please state how long that transgenic lines can survive with P. cinnamomi compared with wild types.
- In line 37, please write the full name of R&D.
- Delete the (Liu et al. 2010) in line 69.
- In line 103, you wrote “Figure 1a, b”. However, in Figure 1, you use capital A and B to show different panels. Please make it consistent.
- In the legend of Figure 1, both abbreviations of secondary embryos and somatic embryos is SSEs. The meaning of the two types of the embryo is not the same. Please revise it.
- In line 105, the full name of SSEs did not show in the front text.
- The result of Kan tolerance is significantly different between ALM80 and TGR3. Please describe more detail in this part.
- The word, carbenicillin, in line 109 should move to line 112 before (300 mg/l).
- The same issue was found in Figure 3. Capital letters were utilized in Fig, but small letters were used in the text. Please make it consistent and all figs in the article.
- The quality of pictures in Figure 6 is really low. Please revised it and check the quality of all pictures in the article.
- In line 313, please add the author name of [32] in the sentence. No subject can be found in this sentence.
- In section 4.1, please write the full name of SH, MS.
- In line 537, Although you have cited the reference for primer sequences in [47], I still suggest that you can present the primer sequences in your supplementary data.
- Delete the (Porth et al. 2005) in line 541.
- What is GD medium in line 614?
Author Response
Dear reviewer,
We are grateful for the helpful corrections and suggestions made by the Reviewer to improve the quality of the original manuscript. Please find our responses to the comments below. The new version of the manuscript has been modified in accordance with the comments.
- In title line 4, is it “thAumatin” ? or it should be Thaumatin. Please check it.
Done.
- In line 11, delete Here.
Done.
- In line 16, what does SE mean? Somatic embryos or secondary embryogenesis?
SE has been changed for somatic embryo.
- Please verify it. If it has mentioned in the front text, please put the abbreviation when you write in the first time.
Done (see line 92).
- In the abstract, please state how long that transgenic lines can survive with P. cinnamomi compared with wild types.
This sentence has been modified as suggested.
- In line 37, please write the full name of R&D.
Done.
- Delete the (Liu et al. 2010) in line 69.
Done.
- In line 103, you wrote “Figure 1a, b”. However, in Figure 1, you use capital A and B to show different panels. Please make it consistent.
In all Figures capital letters has been changed by lowercase letters.
- In the legend of Figure 1, both abbreviations of secondary embryos and somatic embryos is SSEs. The meaning of the two types of the embryo is not the same. Please revise it.
Abbreviations in legend of Figure 1 have been clarified. The abbreviation SSEs corresponds with secondary somatic embryos.
- In line 105, the full name of SSEs did not show in the front text.
Full name of SSEs has been included in the new version.
- The result of Kan tolerance is significantly different between ALM80 and TGR3. Please describe more detail in this part.
This sentence has been modified to address this point and information about the different effect of Kan between ALM80 and TGR3 genotypes has been included in the new version (see lines 91-97).
- The word, carbenicillin, in line 109 should move to line 112 before (300 mg/l).
Done.
- The same issue was found in Figure 3. Capital letters were utilized in Fig, but small letters were used in the text. Please make it consistent and all figs in the article.
See response to suggestion 8.
- The quality of pictures in Figure 6 is really low. Please revised it and check the quality of all pictures in the article.
Pictures in Figure 6 have been changed and quality all pictures has been revised and improved.
- In line 313, please add the author name of [32] in the sentence. No subject can be found in this sentence.
The author name has been included in the new version.
- In section 4.1, please write the full name of SH, MS.
Done.
- In line 537, Although you have cited the reference for primer sequences in [47], I still suggest that you can present the primer sequences in your supplementary data.
We have included this information as Online Resource 3.
- Delete the (Porth et al. 2005) in line 541.
Done.
- What is GD medium in line 614?
GD is Gresshoff and Doy basal medium (see reference 94). This information have been added in the new version.
Thank you again for your effort and do not hesitate to contact again if required.
Sincerely,
Dr. Elena Corredoira
Round 2
Reviewer 1 Report
Dear Authors,
I believe the MS can be published in present form.
Kind regards,
reviewer